# Social Determinants of Breastfeeding Preferences among Black Mothers Living with HIV in Two North American Cities

**DOI:** 10.3390/ijerph17186893

**Published:** 2020-09-21

**Authors:** Josephine Etowa, Egbe Etowa, Hilary Nare, Ikenna Mbagwu, Jean Hannan

**Affiliations:** 1School of Nursing, Faculty of Health Sciences, University of Ottawa, Ottawa, ON K1N 6N5, Canada; imbagwu@uottawa.ca; 2Department of Sociology, Anthropology & Criminology, Faculty of Arts, Humanities & Social Sciences, University of Windsor, Windsor, ON N9B 3P4, Canada; eetowa@uwindsor.ca; 3Nicole Wertheim College of Nursing and Health Sciences, Academic Centre 3, Florida International University, Miami, FL 33199, USA; jean.hannan@fiu.edu

**Keywords:** breastfeeding, black mothers, HIV/AIDs, infant feeding guidelines

## Abstract

The study is motivated by the need to understand the social determinants of breastfeeding attitudes among HIV-positive African, Caribbean, and Black (ACB) mothers. To address the central issue identified in this study, analysis was conducted with datasets from two North American cities, where unique country-specific guidelines complicate infant feeding discourse, decisions, and practices for HIV-positive mothers. These national infant feeding guidelines in Canada and the US present a source of conflict and tension for ACB mothers as they try to navigate the spaces between contradictory cultural expectations and national guidelines. Analyses in this paper were drawn from a broader mixed methods study guided by a community-based participatory research (CBPR) approach to examine infant feeding practices among HIV-positive Black mothers in three countries. The survey were distributed through Qualtrics and SPSS was used for data cleaning and analysis. Results revealed a direct correlation between social determinants and breastfeeding attitude. Country of residence, relatives’ opinion, healthcare providers’ advice and HIV-related stigma had statistically significant association with breastfeeding attitude. While the two countries’ guidelines, which recommend exclusive formula feeding, are cardinal in preventing vertical transmission, they can also be a source of stress. We recommend due consideration of the cultural contexts of women’s lives in infant feeding guidelines, to ensure inclusion of diverse women.

## 1. Introduction

Breastfeeding has been an acceptable method of feeding infants throughout history, not just because it is regarded as being ‘fundamental to womanhood’ and tied to being a good mother [1,2], but also for the extensive benefits it confers on both the mother and the infant [3,4]. In spite of its massive benefits, in the 19th century and early 20th century, the Industrial Revolution and technological advancement aided the creation of feeding bottles and formula feeding, as alternative feeding methods for mothers who were too busy with work and could not breastfeed their babies [5]. Despite the innovation, hygienic challenges and techniques for home storage were generally unsafe and presented some drawbacks [6]. During this period, the low breastfeeding rates in the United States correlated with an increased infant mortality rate. For instance, in Chicago, the infant mortality rate was about 18%, and approximately 53% of babies died from diarrhoea disease. The Chicago Department of Health summarized the estimate as follows: “Fifteen (15) hand-fed babies were dying for every one (1) breast-fed baby” [7].

Realizing that breastmilk was best, the subsequent effect was that the prevalence of breastfeeding increased toward the end of the 20th century, from a breastfeeding initiation rate of less than 25% in the USA by 1971 to more than 60% in 2001 [7,8]. In recent years, the national breastfeeding rates in the United States have shown a relatively upward trend as follows: From a 74.2% breastfeeding rate and 11.9% exclusive breastfeeding rate at 6 months in 2008, to a 83.2% breastfeeding rate and 24.9% exclusive breastfeeding rate at 6 months in 2018. This shows that the United States has already achieved the Healthy People 2010 Target and, according to the 2018 Breastfeeding report, has already met more than half of the Healthy People 2020 Objectives [9]. In Canada, breastfeeding initiation rates have increased significantly from less than 25% in 1965 to approximately 90% in 2015/2016 [10]. In addition, it is worthy to note that significantly more mothers who were of Asian (93.5%) or Black (93.9%) racial background initiated breastfeeding on their last child than did White mothers (86.7%) [11].

As breastfeeding is primarily believed to be the best infant feeding method globally and is culturally embedded in most societies such as the African, Caribbean, and Black (ACB) communities, in some countries, one of the barriers to breastfeeding babies is being diagnosed of Human Immunodeficiency Virus (HIV). This is despite the fact that the WHO recommends that women living with HIV (WLHIV) should breastfeed their babies for at least 12 months and may continue breastfeeding for up to 2 years or longer while adhering fully to antiretroviral therapy (ART) [12]. Some countries have also formulated breastfeeding guidelines with respect to their specific milieu, and, consequently, the implementation of these guidelines varies globally. For instance, western countries like Canada and the US recommend exclusive formula feeding (EFF), while lower-middle-income countries like Nigeria recommend exclusive breastfeeding [13,14,15,16,17,18]. These contradictory implementations of the WHO 2016 guidelines [12] create a lot of tension for Black women, especially those currently living in western countries and who may have also previously lived in developing countries. These tensions are further compounded by the influence of cultural norms on infant feeding and mothering as a whole [19].

However, in the face of these country-specific policies on stipulated infant feeding methods, some WLHIV still choose to breastfeed their babies (rather than exclusively formula feeding them) [1,19,20,21] for fear that their babies could be missing out on the numerous health benefits of breast milk. At the same time, they face possible legal consequences if they infect the baby through breastfeeding and/or if they are caught [22]. Moreover, this decision to breastfeed babies against the guidelines is largely influenced by some factors such as HIV-related personalized stigma, cultural norms, and pressure from other family members, just to mention three. Therefore, there is an urgent need to fully understand the association between these social determinants and breastfeeding preferences (attitudes) among Black WLHIV. This knowledge would enable healthcare providers and policy makers to craft and implement policies that optimize the best infant-feeding practices that would be ideal and safe for both mother and child.

There is also a paucity of data on the prevalence of breastfeeding among Black WLHIV. This study will estimate the prevalence rate and serve as bedrock for future studies on similar subject matters.

## 2. Methods

This research is premised on mixed-method community-based participatory research to examine predictors of breastfeeding attitudes among ACB mothers, living in Ottawa, Canada and in Miami, Florida, US. Documentary analysis laid the foundation for this study, and it then adopted an experimental research design to ensure the measurability of outcomes, to determine the prevalence rate of breastfeeding attitudes and its determinants. The research effectively excavates the nexus between breastfeeding attitudes and various social determinants.

Participants of this research were ACB mothers who are living with HIV, residing in Ottawa, Canada and in Miami, Florida, US. As mothers in this category are a hard-to-reach population, a venue-convenient sampling approach was employed. The women were reached at their event/community centers, and through Aids Service Centers and Healthcare locations. The research survey included in this analysis were Black mothers who had at least one child after HIV diagnosis from Miami (*n* = 201) and Ottawa (*n* = 89). The research was hinged on a venue-based research, where participants had to be drawn from healthcare facilities and community group centers where HIV-positive mothers meet for discussions. The number of participants from the two sites was essentially resourceful to inform the findings of this research; however, it also reflects on the difficulties of recruiting participants of this target group as people do not readily want to divulge their statuses especially to people they do not trust or know.

The scope and procedure of the study was explained to participants, who signed consent forms and voluntarily shared information. The participants could withdraw anytime they felt they no longer wanted to take part in the research willingly, and all protocols were observed for harm reduction to participants.

The outcome or dependent variable for the analysis was breastfeeding attitudes. Breastfeeding attitude was measured using six selected items from the Iowa Infant Feeding Attitude Scale (IIFAS) [23]. The IIFAS did not yield a reliable statistic with our sample, but the selected six items had an acceptable Cronbach’s alpha of 0.76. The six items-modified scale has a benchmark score of 30, where a high score indicates positive attitude toward breastfeeding. They include:Formula-fed babies are more likely to be overfed than are breast-fed babies, IIFS item 5Babies fed breast milk are healthier than babies who are fed formula, IIFS item 9Breast milk is the ideal for babies, IIFS item 12Breast milk is more easily digested than formula, IIFS item 13Breastfeeding is more convenient than formula feeding, IIFS item 15Breast milk is less expensive than formula, IIFS item 16

Participants were tasked with choosing the degree to which they agree with each statement, on a five-point Likert scale ranging from “strongly disagree” to “strongly agree.”

Other parameters included as independent or explanatory variables in the hierarchical linear model were functional social support and personalized HIV stigma. The functional social support (FSS) scale [24] was adapted to rate seven key items with good overall reliability (Cronbach’s alpha = 0.86). The seven items include: I have people who care about what happens to me; I have chances to talk to someone I trust about my health; I have chances to talk to someone I trust about challenges I face as a mother living with HIV; I have chances to talk to someone I trust about challenges I face with feeding my baby as a mother living with HIV; I get invited to go out and do things with other people including mothers living with HIV; I get useful advice about things that are important to me as a mother living with HIV; I get help when I am sick in bed. Responses were based on a five-point Likert-type scale of: Much less than I would like = 1, less than I would like = 2, some but would like more = 3, almost as much as I would like = 4, as much as I would like = 5. Hence, the maximum score attainable on the scale was 35.

The first three items on the revised HIV stigma scale [25,26,27] captured personalized HIV stigma. The items adapted include the following: I have been hurt by how people reacted to learning I have HIV, and I have stopped socializing with some people because of their reactions to me having HIV. The responses were adapted to ‘yes’ (1 point) or ‘no’ (0 points). Hence, total scores per respondent ranged from 1 to 3 points.

To measure other independent variables (other social factors potentially associated with breastfeeding attitudes), we asked questions about the participant’s awareness of the infant feeding guidelines in their city of residency, what was their infant’s father’s/spouse’s, family members’, and healthcare provider’s opinion about breastfeeding their baby. The mothers were also asked if their cultural beliefs contradicted with the infant feeding guidelines.

Data collection occurred between November 2016 and March 2018. Surveys were provided online and hard copies were also provided for those who had hardships in accessing the copies online, due to a number reasons including among them a lack of technological competencies. The survey explored breastfeeding attitudes among ACB mothers in relation to national guidelines of their city of residency. Participants completed a demographic questionnaire to assess breastfeeding attitudes. Participants were asked about the number of people in their household, number of children after HIV-positive results, number of years since diagnosis of HIV, education, marital status, employment status, etc. Values attained from the two cities were analyzed on a comparative basis.

Qualtrics software, Copyright © (2016–2018) was used in this study, and, in instances where it was not feasible, we resorted to paper-and-pencil surveys that were later entered into IBM SPSS Statistics (SPSS). Data directly entered into Qualtrics were downloaded into SPSS for data cleaning and analysis. Paper-and-pencil data were double-entered into SPSS to ensure accuracy of data for analysis. Data from both sites were then entered into SPSS, cleaned, and merged into a single aggregated file for analysis. Data entry, cleaning, coding, and analyses commenced in 2018, after data collection came to an end.

This manuscript is based on our original three cities in three countries study funded by the Canadian Institutes of Health Research (CIHR), Funding Reference (FRN): 144831. The study received ethics approved from the Health Sciences and Science Research Ethics Board at the University of Ottawa (certificate H08-16-27) approval date 12/08/2016, renewal dates 12/08/2017 and 12/08/2018, the Carleton University Research Ethics Board-A (CUREB-A, certificate 106300), the Social and Behavioral Institutional Review Board at Florida International University (certificate 105160), and the Research Ethics Committee at the University of Port Harcourt (certificate UPH/CEREMAD/REC/04).

The descriptive statistics analyses of variables (Table 1 and Table 2) were preceded by hierarchical linear modeling (HLM) to determine the association between the predictor variables and the outcome variable (breastfeeding attitude). HLM was appropriate to control for the possible effects of geographic clustering of our samples (Ottawa versus Miami) alongside those socio-demographic factors. Such control enables the separation of the effects of key parameters of interest such as the influence of social networks, health providers, and social support on breastfeeding attitude. Hence, possible bias parameter estimates are avoided using the hierarchical linear modeling. Two blocks of variables were entered into the model in a two-step analysis. In the first step, block 1, the control variables, which include the city of residence and sociodemographic variables, were entered into yield model 1 in Table 3. Next, block 2 or key parameters of interest (predictor variables) were added to produce parameter estimates for model 2. Model 2 became the lead model for results interpretation while comparing changes in R-squared from model 1 to model 2.

## 3. Results

This section presents results of HLM to predict breastfeeding attitudes of Black mothers living with HIV in Ottawa, Canada and Miami, Florida, US. Prior to these, Table 1 and Table 2 present the descriptive statistics of the data employed in the analysis.

Participants from both sites had been diagnosed with HIV for more than a decade with the Canadian site having 12.7 ± 6.4 years, while the US site participants had a range of 10.9 ± 7.3 years. In both sites, mothers had babies after their HIV+ results. Employment rate for these mothers was higher in the Canadian site than in the US site, translating to 57% against 32.7% accordingly. Mothers in both countries had babies after being diagnosed with HIV; on average, mothers in Ottawa had two more children after their seropositive status, while in the Miami site, mothers had just one child after the serostatus result. In Ottawa, family members per household were higher than those in Miami, as the average number was four and three, respectively.

In terms of relationships and marital status, 66.5% and 37.5% were not married in Ottawa, Canada and Miami, US; respectively. They were not married either because they were separated, divorced, or widowed. Of the Canadian participants, only 33.3% were married, while 60.8% in the US were married.

More mothers were employed in Ottawa, Canada than in Miami, US. Specifically, 57.3% of mothers from the Canadian site were employed compared to just 32.7% in the US site; inversely, it implied that 42.7% of mothers in Ottawa were unemployed, while 67.3% were unemployed in Miami.

Generally, education levels were quite high in both countries with 40% of the mothers in Ottawa having attended high school, or technical or vocational school, while 65.8% had done the same in Miami. Others had attended college or university education with participants in Canada constituting 58.8% of the total population, and with those of Miami, US, 33.2% had attended some college or university. Only one participant in Ottawa had some primary school education.

Results from this study showed that breastfeeding attitudes were higher in Ottawa than in Miami. Predictor variables were assigned scores and breastfeeding attitudes were measured on a scale of 1 to 30, Max = 30 (M ± SD). In the Canadian site, breastfeeding attitudes emerged to be 24.70 ± 4.50, while in the US site, it was 20.89 ± 4.70. In terms of social support, it was 24.79 ± 6.52 and 21.81 ± 8.60, respectively.

Social support was also assigned scores, Max = 35 (M ± SD). In that respect, in Ottawa, Canada, functional support scores were higher than those in Miami, US, measuring 24.79 ± 6.52 in Canada against 21.81 ± 8.60. HIV personalized stigma score was paged at 3, Max = 3 (Median, Mode). Stigma scores were 1 in Ottawa and 2.3 in Miami.

Opinions on healthcare provider’s opinions were also assessed and measured in both countries. Specifically, 100% of the ACB mothers in Ottawa, Canada valued healthcare providers opinions, while, in Miami, US, figures dropped a bit to 84.6% of all the mothers who took part in this study. Last but not least, spouse, partner, and baby father’s opinion was rated to check how mothers valued such opinions, and in the Canadian site, 71.3% valued such opinions, while it was important for only 53.7% of mothers in the US site.

The results of hierarchical linear modeling (Table 3) show that after controlling for socio-demographic characteristics, social factors including spousal, family, and health providers influences alongside social support and years since HIV diagnosis contributed to a 19% (0.38–0.19) variation in breastfeeding attitudes score of the Black mothers living with HIV at a statistically significant level (*p* < 0.01). The results show that socio-demographic factors including city of residence (R^2^ = 0.19), and the social factors alongside years since HIV diagnosis (R^2^ = 0.38–0.19 = 0.19), were influencing breastfeeding attitudes at the same rate (*p* < 0.01).

Based on results on model 2 in Table 3, specific influences of the geographic and social factors on breastfeeding attitudes include:Breastfeeding attitudes score was greater in mothers residing in Canada than those in the US (β = 0.41) at *p* < 0.01Increased rated scores of relatives’ opinion associated with increased breastfeeding attitude score (β = 0.38) at *p* < 0.01Increased rated scores of health providers’ opinion associated with increased breastfeeding attitude score (β = 0.19) at *p* < 0.05Increased stigma score associated with increased breastfeeding attitude score (β = 0.15) at *p* < 0.05

## 4. Discussion

### 4.1. Residing in Ottawa, Canada versus Miami-Florida, US

This study revealed that city of residence had an important bearing on breastfeeding attitudes. For instance, Britain has stipulated criteria for WLHIV who choose to breastfeed their babies. However, if any of the criteria is flouted, the woman would be referred to Social Care as the child is at significant risk of HIV infection [28]. Thus, in Britain, mothers can choose an infant feeding method of their choice, subject to strictly adhering to a set of guidelines. This is not the case in the US and Canada, where there are no such criteria for WLHIV, as mothers are not afforded the chance to make choices, but rather, they must adhere exclusively to formula feeding, otherwise breastfeeding may have legal consequences. In the US, there are multiple instances where some WLHIV lost custody of their children for failing to adhere to the ‘no breastfeeding’ guideline [7,29], while in Canada, the guidelines are not always clear [22] and criminal charges in such circumstances seem unlikely because it is not generally in the best interest of the child [30].

These factors seemed to have influence on breastfeeding attitudes as shown by results of our study. Our findings showed that breastfeeding attitudes scores were greater in mothers residing in Canada than those in the US (β = 0.41). Prosecutions for violating infant feeding guidelines were stronger in America, which could be the reason why there was a difference, despite the two countries having similar national guidelines on the subject. In regions where breastfeeding was permissible as per the WHO recommendations, breastfeeding attitudes were even higher, compared to both US and Canada, which had restrictive national guidelines. For instance, in a cross-sectional study among 600 HIV-positive mothers in prevention of vertical transmission (PMTCT) clinics in Southwestern Nigeria, the exclusive breastfeeding rate was about 61.0% among mothers who had children of less than 7 months of age [31]. This rate is analogous to those obtained from studies in other African countries: 85.5% in Ethiopia [32], 80.4% in Kenya [33], 89% in Lesotho [34]. Thus, a critical analysis of all this information indicates that city of residence had a bearing on breastfeeding attitudes.

### 4.2. Stigma

Numerous literatures have been published on the purported association between HIV-related personalized stigma and breastfeeding attitude and practice [1,31,35,36,37,38]. In a cross-sectional study among 550 WLHIV in Southeast Nigeria, approximately two-thirds (64.7%) of the study participants knew that HIV could be transmitted through breastmilk, yet some of them persisted with breastfeeding their babies (17.6%) for fear of their HIV status being disclosed (*p* < 0.02) [37]. Similarly, in a Canadian qualitative study that involved narrative interviews with HIV-positive mothers in Ontario, some of the participants were concerned that if they do not breastfeed their babies, it would lead to undesired divulgence of their HIV status to people [38]. Hence, stigma has an important bearing on decision-making as far as infant feeding choices are concerned.

In our study, the participants also shared similar sentiments showing that stigma had an important influence on breastfeeding attitudes. Our findings revealed that increased stigma score was associated with increased breastfeeding attitude score (β = 0.15). Thus, some mothers are cornered into breastfeeding to protect their reputations, image, and dignity despite the recommendations not to breastfeed in these two North American countries. Many women living with HIV fear unwanted disclosure of their HIV status, as this can have long-term emotional, financial, and health consequences for them such as suicidal thoughts, inability to find work, and non-adherence to medications, just to mention three [38,39,40,41].

### 4.3. Healthcare Providers

Healthcare workers’ opinion has significant statistical association with breastfeeding attitudes. Exclusive breastfeeding occurred when a woman was motivated regarding motherhood, had correct learning and understanding about infant feeding practices through counselling, no fear of breastfeeding or impact of opposing feeding-related cultural belief, and the support from others to be assertive about their feeding choices when faced with pressure to mix feed [42]. Thus, mothers had so much faith in healthcare facilities and believed the advice of healthcare workers to be trustworthy as they are professionals. With the full support and backing of these professionals, breastfeeding attitudes tended to be high. This is also similar to the findings of our research, which showed that increased rated scores of health providers’ opinion were associated with increased breastfeeding attitude scores (β = 0.19).

A study conducted by Koricho et al. [43] showed how powerful healthcare workers’ opinions can be in influencing breastfeeding attitudes. Their findings showed that, “Health care workers impacted on infant feeding choices of mothers by instilling fear and the message came across so strong that mothers felt like breastfeeding was poisonous” [43]. Thus, mothers felt very guilty to breastfeed and, those who did, did so with guilty conscience and had no choice, as they could not afford formula feeding. Thus, healthcare workers’ opinion is critical in determining breastfeeding attitudes, and attitudes tended to be higher where mothers had blessings from their healthcare workers.

### 4.4. Spouse Support

Several authors have documented the association between spousal or baby’s father support and positive breastfeeding attitude and intentions [44,45,46,47,48]. Correspondingly, among WLHIV, there seems to be a similar association between both variables, as women who succeeded in breastfeeding had the unequivocal support of their husbands or male partners [35,36]. In a hospital-based cross-sectional study among 600 HIV-positive mothers in Southwest Nigeria, a large proportion (84.0%) of the study participants chose to exclusively breastfeed their babies because of their spouses’ influence [31].

In a study in the US that involved a convenient sample of 112 pregnant women and their male partners, higher scores on IIFAS for both mothers and partners had a statistically significant association with their intentions to breastfeed. With each point increase in IIFAS score, the odds that the mother intended to breastfeed in the first few weeks increased 19–20% (OR = 1.19, 95% CI = 1.08–1.30 for partner’s IIFAS score; OR = 1.20, 95% CI = 1.10–1.32 for mother’s IIFAS score) [49].

### 4.5. Relatives

Breastfeeding attitudes and intentions among women are significantly influenced by the opinions of the other members of the family, and this has been studied extensively [31,35,37,50,51,52]. These family members include the grandmothers of the infants, and aunts and sisters of the mothers, and their influence could cause undue pressure on the mother [50]. In a study conducted in Tanzania among 446 HIV-positive mothers to ascertain the role of mothers-in-law in the PMCTC of HIV, it was noted that mothers-in-law expected their daughters-in-law to breastfeed in a culturally accepted manner, and were generally negative towards the infant feeding methods recommended for WLHIV [53].

These research studies are congruent with findings of our study. We obtained that increased rated scores of relatives’ opinions are associated with increased breastfeeding attitude scores (β = 0.38). It is pertinent to note that this pressure from other family members could eventually make some WLHIV cave in to the excessive demands and breastfeed their babies, even though they are aware of country-specific guidelines on infant feeding and the increased risk of vertical transmission of HIV when they breastfeed. Therefore, health programs aiming to improve healthy infant feeding practices in Canada and the US should consider strategies to address these pressures including involving other family members such as grandmothers in their interventions.

In summary, after a careful and detailed analysis, this research has numerically shown the relationship between social variables and breastfeeding attitudes. It has detailed the perspectives of mothers and provided the prevalence rate of social determinants of breastfeeding among mothers. Proper procedures and instruments such as Qualtrics and SPSS were used to ensure the accuracy and trustworthiness of the results to provide a realistic explanation of the phenomena under study. After a critical analysis of social determinants of breastfeeding attitudes or preferences among Black mothers living with HIV, the testimonial “of good motherhood” entails feeding and caring for babies, providing all the necessary nutrients for growth [1,2,3]. Primarily, breastfeeding is often viewed as the default infant feeding method and it is an experience highly valued especially by ACB as its roots are deeply entrenched in cultural norms. Despite breastmilk value and massive well-known benefits, HIV and AIDS have altered the breastfeeding discourse, especially in North American countries where it is not recommended for seropositive mothers. This leads to stress and pressure as the mothers try to navigate the spaces between their cultural expectations and national infant feeding guidelines. It is imperative that healthcare providers and policy makers apply empathy while dealing with WLHIV who choose to breastfeed. For instance, if a provider takes a harsh stance against breastfeeding, the woman may shy away from discussing her infant feeding worries and may eventually breastfeed in secret. This would make the provider miss the rare opportunity to adequately counsel the woman in a way that would help her make the best decisions for her baby’s health and wellbeing [19,28].

### 4.6. Recommendations

This paper strongly recommends a policy development and implementation process that ensures meaningful engagement of those affected by the policy, in this case, Black women living with HIV. As an indigenous saying goes, ‘nothing about us is for us without us.’ This approach is to promote culturally responsive infant feeding practices in both the US and Canada. Evidence-based research has shown that despite the strict guidelines in both countries, some mothers end up violating these guidelines in secrecy, thus putting the baby at risk [1,19,20,21]. Henceforth, creating the much-needed openness and enabling environment for those willing to breastfeed will help create systems for monitoring and evaluation to ensure mothers are adhering to the mechanisms than live a lie. Thus, an improvement in policy will go a long way to preventing vertical transmission.

Closely related to the issue of policy is the need to carefully examine healthcare providers’ perspectives on infant feeding among mothers living with HIV, and to rejuvenate counselling modules to build the capacity of providers so that they can be better supported to provide high-quality perinatal care for HIV-positive Black mothers. These include the care of those who may choose to breastfeed their infant, which, at this time, is a taboo or criminal activity in the North American context. Their support and provision of required information on best practices when one chooses to breastfeed become vital in reducing PMTCTs.

There is a need to increase public awareness about HIV and AIDS including the advancement in its treatment. The public needs to understand that HIV is no longer a death sentence. A better understanding of the disease will increase support for those affected by HIV/AIDS and reduce commonly held stereotypes. This will also go a long way in enhancing social support and reducing stigma. Moreover, while culture is very important in one’s life experiences, some cultural expectations and beliefs may not result in the intended positive outcomes as anticipated. Hence, there is a need for critical health literacy in the Black community to foster better understandings of HIV/AIDS and promising practices for infant feeding, especially in the context of seropositive status. For instance, although mixed feeding may be culturally viewed as important and encouraged by friends and family members [54] who may believe that breast milk is not enough on its own, regardless of the risk it poses to the baby, there is evidence that mixed feeding poses more risk of vertical transmission of HIV than exclusive breastfeeding [55,56]. Mixed feeding during the first six months of the baby’s life has a higher risk of HIV transmission because the other foods given to the baby concomitantly with breastmilk can damage the already delicate and porous intestinal walls, and this may allow the virus to be transmitted more easily [56]. This kind of information needs to be integrated into public education of Black community members in order to improve health literacy and HIV/AIDS, and may help in preventing vertical transmission. Lastly, countries should strive to adopt the WHO recommendations so that if one person relocates from one country to another anywhere in the world, infant feeding guidelines will remain the same, and mothers would have options to consider their decisions about infant feeding. This will standardize guidelines across countries and may eliminate much of the problems outlined in this paper.

## 5. Conclusions

There are several social determinants that influence breastfeeding attitudes among Black mothers. While the Canadian and US national guidelines, which recommend exclusive formula feeding, are cardinal in prevention of vertical transmission (PMTCT), they can also be a source of enormous emotional stress and suffering to HIV-positive mothers as unintended negative consequences of these policies. The internal tension created by the reality of mothers—who find themselves living within the spaces of cultural expectations of breastfeeding as an expression of ‘good motherhood’—and the need to abide by national infant feeding guidelines require attention by health and social service providers, as well as policy makers. This is particularly problematic for ACB mothers who relocate from low-resource countries such as Nigeria, where they could breastfeed despite their HIV status, to a western nation like Canada or US where exclusive formula feeding is the norm. Policy making needs to consider these important variables, as this has a bearing on infant feeding, as well as their health. The quandary for many WLHIV in Canada and US seems to be how to breastfeed their babies without facing the repercussions for flouting the guidelines. Current demographics in both the US and Canada show diversity; hence, there is a need to implement culturally responsive polices, especially in the areas of maternal–infant healthcare including motherhood.

## Figures and Tables

**Table 1 ijerph-17-06893-t001:** Socio-demographic characteristics.

Characteristics	Ottawa	Miami
	*n* (%)	*n* (%)
Number of participants (N)	89	201
Mothers age, (M ± SD)	36.6 ± 6.4	32.4 ± 5.8
Number of persons in household, Median (Range)	4 (1–7)	3 (1–8)
HIV-related information:		
Number of children born after HIV+, Median (Range)	2 (1–3)	1 (1–3)
Number of years since HIV+, (M ± SD)	12.7 ± 6.4	10.9 ± 7.3
Education:		
Primary school	1 (1.2)	0 (0.0)
High school, or technical or vocational school	34 (40.0)	131 (65.8)
College or university	50 (58.8)	66 (33.2)
Relationship status:		
Single/separated/divorced/widowed	57 (66.5)	61 (35.7)
Married	29 (33.3)	121 (60.8)
Employment status:		
Employed (full time or part time)	51 (57.3)	65 (32.7)
Unemployed	38 (42.7)	134 (67.3)

**Table 2 ijerph-17-06893-t002:** Descriptive statistics of the outcome (breastfeeding attitudes) and other key predictor variables.

Response Categories	Canada	USA
	*n* (%)	*n* (%)
Breastfeeding attitude score, Max = 30 (M ± SD)	24.70 ± 4.50	20.89 ± 4.70
Functional social support score, Max = 35 (M ± SD)	24.79 ± 6.52	21.81 ± 8.60
HIV personalized stigma score, Max = 3 (Median, Mode)	1, 0	2, 3
Spouse/partner/baby’s father’s opinion rated very important or important	62 (71.3)	108 (53.7)
Cared very much or cared about the other family members/close relatives’ opinion	66 (74.2)	97 (48.3)
Cared very much or cared about the health provider’s opinion	89 (100)	170 (84.6)

**Table 3 ijerph-17-06893-t003:** Summary of hierarchical linear regression predicting breastfeeding attitude.

Independent Variables	Model 1			Model 2		
Variable	B	Std. Error	β	B	Std. Error	β
City of residence (Ottawa, Canada = 1, Miami, US = 0)	4.43	0.88	0.41 **	2.67	0.98	0.25 **
Household size (persons)	0.04	0.06	0.06	0.05	0.06	0.07
Formal education (years)	0.47	0.27	0.14	0.40	0.25	0.12
Marital status (married = 1, otherwise = 0)	0.04	0.88	<0.00	0.18	0.80	0.02
Rating of spouse/partner/baby’s father’s opinion on infant feeding				−0.24	0.12	−0.17
Rating of other relatives’ opinion				0.41	0.10	0.38 **
Rating of health providers’ opinion				0.39	0.18	0.19 *
HIV status Disclosure (ranked score)				−0.10	0.68	−0.12
Personalized HIV stigma (scale score)				0.63	0.32	0.15 *
Functional Social Support (scale score)				0.05	0.06	0.07
Years since HIV status diagnosis				<0.01	0.06	<0.01
*R* ^2^		0.19			0.38	
F for ∆*R*^2^		7.95 **			5.24 **	

* *p* < 0.05, ** *p* < 0.01, B = unstandardized beta coefficient, β = standardized beta coefficient.

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
