# Peer review of "Social Determinants of Breastfeeding Preferences among Black Mothers Living with HIV in Two North American Cities"

_ijerph, 2020, doi:10.3390/ijerph17186893_

Round 1
Reviewer 1 Report
The article described a research study to determine social determinants of breastfeeding attitudes among African Caribbean and Black (ACB) mothers who are living with HIV. The study was driven by the number of African Caribbean and Black mothers living with HIV violate the United States and Canada breastfeeding guideline for women living with HIV (WLHIV). Based on the study findings, the authors called for consideration of cultural contexts of WLHIV who breastfeed their infant to include diverse women as well as educating ACB mothers who are living with HIV.
Introduction Section:
- The arguments on the benefits of breastfeeding to infants were solid. However, without any explanation, the authors introduced the US and Canada breastfeeding guidelines for mothers living with HIV, which recommend (or prohibit?) WLHIV to breastfeed their infants. In the abstract, the authors made mention of the cardinal in preventing vertical transmission of HIV. Still, there was no mention of such in the introduction section. Perhaps, some explanation and statistics would improve the section.
- The authors need to be consistent with the choice of word regarding the breastfeeding guideline (or law/regulation?) for mothers living with HIV. The terms recommended and prohibit had two different meanings and resulted in 2 different outcomes. Guideline or law/regulation?
- Also, the last sentence in the introduction section indicated that the authors attempted to determine the prevalence of breastfeeding among Black (or ACB) mothers living with HIV. The choice of the word determine seemed misleading since data collected from 2 cities would give an estimate.
- On page 2, 2nd paragraph, the authors needed to spell out what ART
Methods Section:
- The authors repeatedly mentioned that the study’s participants resided in the US and Canada, which might give readers the wrong impression that data was drawn from national samples. Perhaps, they would reword to Miami, US, and Ottawa, Canada.
- The authors detailed how participants were selected but did not specify as if the samples were convenient ones, which included mothers living with HIV who resided in Miami and Ottawa and enrolled in the study.
- The authors also described the study instruments in detailed, yet did not specify the measures, such as independent/predictive, dependent/outcome, and control variables. I assumed that the breastfeeding attitude score was the outcome variable; however, its descriptive statistics were included in the table of descriptive statistics of the predictive variables.
- There was no mentioning of data analysis in the method section.
Results Section:
- As stated above, the authors did not mention the data analysis; still, in this section, they presented the results of hierarchical linear regression.
- In table 3, why did the authors have two models? What was B and b?
- Since the study data were cross-sectional, the authors should choose the word associated rather than
- The authors’ interpretation of the result as country of residence was troublesome since data was based on convenient samples from 2 cities. As aforementioned, this direction of interpretation was misleading to the readers.
Discussion Section:
- The authors began this section stating that This study revealed that country of residence had an important bearing on breastfeeding attitudes, then presented literature information from Britain and the US.
- Convenient samples from two cities should not be interpreted as representing countries.
- I could not connect this study’s findings with the authors’ recommendation of a change in policy framework to enable culturally responsive infant feeding practices in the US and Canada.
Conclusion Section:
- The authors should make the distinction between law or guidelines/recommendations of breastfeeding for mothers living with HIV.
- The authors should not generalize the results.
Author Response
Please, find attached the table that contains a point-by-point response to the reviewer's comments.

Reviewer 2 Report
This is an excellent paper and valuable content.
There are some points of clarification I will recommend:
- I discourage any reference to 'breast is best.' This is an outdated slogan that causes immediate strife. We do know that human milk is optimal nutrition for human babies and breastfeeding provides the ideal feeding method for neurological and skeletal/muscular development. However, 'breast is best' is tied very closely to the formula debate and may distract from your intention with the paper.
- As indicated, WHO does recommend breastfeeding for WLHIV when adhering fully to ART. Was ART adherence measured for this population? Did that influence breastfeeding preferences?
- Being a Mother Scale was used to measure mothers' experiences but the results are not identified. How did these results factor into the study populations' breastfeeding experience?
- "Functional social support scores": The only 2 places this appears in the manuscript are Tables 2 and 3. Please indicate in the methods section how this is measured.
- Please be consistent when referring to the U.S. All of the following are used throughout the manuscript: US, USA, the United States, America.
- Under "Recommendations": Please provide a reference for the statement regarding mixed feeding, "...there is evidence that poses more risk of vertical transmission of HIV."
Author Response
Please, find attached the point-by-point response to the reviewer.
Thanks.

Round 2
Reviewer 1 Report
Congratulations!